# Parameter optimization for epidemiological SEPAI3R3O model with genetic algorithm

## Abstract

In this study, we propose a variation of the SEIR epidemiological model, called SEPAI3R3O, and apply genetic algorithms to analyze and optimize the associated parameters. This model was developed based on the analysis of sociodemographic and behavioral data from anomalous ICDs (International Classification of Disease) and ICPCs (International Classification of Primary Care) collected from units specialized in SARS (Severe Acute Respiratory Syndrome)(i.e., specifically flu and COVID-19) in the city of Recife, located in northeast Brazil, from April $26, 2020$, to March $7, 2021$. The main objective was to understand the dynamics of disease spread and identify critical factors that influence their spread. One of these factors is the underreporting rate, estimated at around $50\%$, which significantly increases cases due to inadequate testing. We could precisely adjust the model parameters using a genetic optimization approach, resulting in more accurate disease dynamics predictions and a more realistic view of the number of people infected by SARS. The results indicate that the SEPAI3R3O model, when optimized with genetic algorithms, could predict the spread of the disease with an effective reproduction rate $R_0$ of $3(95\%$ CI 2.8–3.2$)$ and a growth rate of $0.014(95\%$ CI 0.013–0.015$)$ for the period analyzed. With realistic data, this approach offers a valuable tool for researchers and healthcare professionals in making decisions and formulating more effective intervention strategies.

## 1 Introduction

The search for optimization and analysis of complex systems is a constantly highlighted area of research, with particular relevance when modeling the spread of infectious diseases Anderson et al. (2020). The need for accurate and efficient models has become more critical in the current context, where new diseases emerge and existing pathogens evolve. Among the tools that have stood out in this scenario, genetic algorithms, a subarea of evolutionary computing, have demonstrated success in several complex problems, from design optimization to financial predictions Goldberg (1989). These algorithms offer a robust and adaptable approach to understanding and anticipating disease dynamics in epidemiology.

The proposed model is an example of innovation in this area, inspired by the widely adopted SEIR model, used to characterize historically relevant infectious diseases such as the Spanish flu Organization (2020). SEPAI3R3O was developed to analyze sociodemographic and behavioral data from ICD and ICPC collected from specialized health units. This model provides a detailed view of disease progression and individual interactions, making it a valuable tool for researchers and healthcare professionals. Integrating genetic algorithms into this model optimizes the analysis of disease spread and enables a deeper understanding of the factors that influence spread Mitchell (1998).

The increasing complexity of healthcare systems and the need for rapid responses in outbreak situations highlight the importance of combining genetic algorithms with detailed epidemiological models, such as SEPAI3R3O, as a strategic key to predicting and mitigating future disease outbreaks.

In this context, it is essential to address the challenge of finding the correct parameters for dynamic epidemic models. The predictive success of these models is directly related to the adequate adjustment of the training data, avoiding both underfitting and overfitting. When training data is insufficiently fit, the model may diverge or produce overestimates with high variance. On the other

hand, overfitting, a common problem in epidemic models, occurs when model parameters are excessively influenced by training data, resulting in low-variance but unrealistic predictions Yang et al. (2020a).

Overfitting remains a significant challenge, especially in dynamic epidemic models, due to the fluctuation of several parameters within their uncertainty ranges Basu & Andrews (2013). To mitigate this problem, some epidemic analyses, including studies on outbreaks such as COVID-19, have applied restrictions to reduce the number of free parameters, thus controlling overfitting and preserving the relevance of these studies Peng et al. (2020).

The remainder of this article is organized as follows: Section 2, models and methods, describes the SEPAI3R3O model. Section 3 provides a detailed review of genetic algorithms and their application in epidemiological modeling. Section 4 presents the results and discussions. Section 5 addresses conclusions.

## 2 Models and Methods

Modeling complex systems, especially in epidemiology, requires a robust and adaptable approach. In this study, we combine the effectiveness of genetic algorithms with the accuracy of the proposed SEPAI3R3O epidemiological model to analyze the spread of diseases. This section details the methodology adopted, describing the compartmentalized model and the implementation of the genetic algorithm.

### 2.1 Compartmented Model: SEPAI3R3O

The use of compartmental models to study infectious diseases has been massively used in the literature Tuite et al. (2020); Labadin & Hong (2020); Wu et al. (2020b). The SEIR model, for example, divides the population into susceptible, exposed, infectious, and recovered compartments. Each of these compartments was modeled using a set of differential equations Anderson & May (1992).

The SEIR model traces the progression of individuals through stages of a disease: Susceptible (S), Exposed (E), Infected (I), and Recovered (R). Transitions between these groups are determined by entry and exit rates. Specifically, as individuals leave one stage, they enter the next; for instance, those exiting the 'susceptible' stage enter the 'exposed' category. The model assumes that once recovered, individuals cannot be reinfected. The speed of disease spread is influenced by the contact between infected and susceptible people, defining the rate at which susceptibles become exposed. Once exposed, they aren't immediately contagious, but after an incubation period, they become infectious. The infected group's exit rate is tied to either recovery or death, with the death rate often represented by the case fatality rate (CFR). The primary reproduction number (R0) quantifies the contagion's effectiveness, indicating how many others an infected individual can potentially infect. Meloni et al. (2011).

In this article, we will apply a variation of this model that we call SEPAI3R3O. The increased number of stages in each compartment indicates that the groups were subdivided into more subgroups. That is, in this model, we used Exposed (E), Pre-symptomatic (P), Asymptomatic (A), three groups of infected (I), and three groups of removed (R), in addition to adding a new group called O, which represents the dead. Separating the dead as a new group is a practice used in other adapted models since recovery and death are opposite consequences for those infected, in addition to the need for analytical interest in both cases.

The model includes the possibility that exposed individuals who have not yet developed symptoms can transmit the virus ("pre-symptomatic transmission"). To model this situation, we added class P (no symptoms but can transmit). Another common condition is the presence of people who contract the virus but do not show symptoms, and for this reason, class A (asymptomatic people) was included. The three groups of infected people were created to represent three distinct phases of infection: $I_1$ infected people, but with mild symptoms $I_2$ people infected and tested for showing symptoms of medium severity (for which they can even be hospitalized) $I_3$ people infected, tested and showing symptoms of high severity Following the same logic as the SEIR model, for a person to be hospitalized in a severe $I_3$ condition, they must necessarily have gone through stage P, infectious, but without symptoms and testing, and then through stage $I_2$, with enough symptoms to

require testing. It only enters stage $I_3$ if the condition becomes more serious, requiring intensive treatment in a hospital unit. Using this premise, only people who are in group $I_3$ can die, passing to group O; that is, in this model, the path taken for an individual to die is S→P→$I_2$→$I_3$→O or S→P→$I_1$→$I_2$→$I_3$→O.

For the individual to be able to migrate to the group of recovered people (R), they must first be in one of the groups of people in recovery ($R_1$, $R_2$, and $R3$), which represent individuals who are recovering from the disease in the various stages of the disease intensity (i.e., groups $I_1$, $I_2$ and $I_3$ respectively). The motivation behind creating such groups was the need to consider recovery time depending on the severity of the infection, which allows us to model the demand for hospital beds satisfactorily. Therefore, groups $R_1$, $R_2$, and $R_3$ do not communicate; that is, there are four distinct paths for recovery (R) of an infected person, being: $I_1 \rightarrow R_1 \rightarrow R$ (i.e., faster recovery time); $I_1 \rightarrow I_2 \rightarrow R_2 \rightarrow R$ (i.e., recovery with intermediate time); $I_1 \rightarrow I_2 \rightarrow I_3 \rightarrow R_3 \rightarrow R$ (i.e., longer recovery time).

In addition to the addition of subgroups, shown previously, this model was designed to consider the underreporting rate ($P_{sub}$) of the number of cases, as, with the division of subgroups, there is the possibility of controlling the proportion of people who migrate from $I_1$ to $I_2$ (i.e., cases that present sufficient symptoms for testing), or that migrate from $I_1$ to $R_1$. Thus, these individuals represent cases of underreporting, as they contracted the disease but were never tested. Below are the ordinary differential equations of the model:

$$\frac{dS}{dt} = -P_{sub} \times (P + A + I_1 + I_2 + I_3 + O)S \tag{1}$$

$$\frac{dE}{dt} = (\beta_e P + \beta_0 A + \beta_1 I_1 + \beta_2 I_2 + \beta_3 I_3) \times S - a_0 E \tag{2}$$

$$\frac{dP}{dt} = a_0 \times E - a_1 P \tag{3}$$

$$\frac{dA}{dt} = f_{a1} \times P - c_0 A \tag{4}$$

$$\frac{dI_1}{dt} = p_0 \times ((1 - f) \times a_1 \times P) - (c_1 + p_1) \times I_1 \tag{5}$$

$$\frac{dI_2}{dt} = p_1 \times I_1 - (c_2 + p_2) \times I_2 \tag{6}$$

$$\frac{dI_3}{dt} = p_2 \times I_2 - (c_3 + \mu) \times I_3 \tag{7}$$

$$\frac{dR_1}{dt} = (d_0 + c_0) \times I_1 \tag{8}$$

$$\frac{dR_2}{dt} = (d_1 + c_1) \times I_2 \tag{9}$$

$$\frac{dR_3}{dt} = cf - (d_2 + c_3) \times I_3 \tag{10}$$

$$\frac{dR}{dt} = t_0 \times R_0 + t_1 \times R_1 + t_2 \times R_2 \tag{11}$$

$$\frac{dO}{dt} = cf \times I_3 \tag{12}$$

Where:

- $\beta_i$ - Rate at which individuals infected in class $I_i$, where ($i = 1, 2, 3$), come into contact with susceptible individuals and infect them.
- $a_i$ - Class progression rate from exposed to infected;
- $f$ - Fraction of all asymptomatic infections

The parameters $\beta_i$, ($i = e, 0, 1, 2, 3$) represent the transmission rates in the different phases of the disease, while the parameters $c_i$ ($i = 0, 1, 2, 3$) represent the different rates of recovery and $\mu$ the mortality rate. The parameters $a_i$ ($i = 0, 1$) indicate the exit rate of classes E and P, respectively. The

parameters $p_0$, $p_1$, and $p_2$ are, respectively, the rate of progression from mild to severe infection and from severe to critical infection. Because these rate constants are generally not measured directly in studies, they are related to clinical observations using the following formulas:

- $P_{sub}$ - Subreporting rate
- $P_{I_3}$ - "Proportion of cases arriving at $I_2$ that go to $I_3$"
- $a_0$ - Exposed rate without symptoms or transmission $E = 1/PresymPeriod$
- $a_1$ - Pre-asymptomatic rate = $1/(\text{IncubPeriod} - \text{PresymPeriod})$
- $p_0$ - Transition rate from $I_1$ to $I_2 = (1 - P_{sub}) \times 1/tI_1$
- $p_1$ - Transition rate from $I_2$ to $I_3 = PI_3 \times 1/tI_2$
- $p_2$ - Transition rate from $I_3$ to $O = c_f \times 1/tI_3$
- $c_0$ - Transition rate from $I_1$ to $R_1 = P_{sub} \times 1/tI_1$
- $c_1$ - Transition rate from $I_2$ to $R_2 = (1 - PI_3) \times 1/tI_2$
- $c_2$ - Transition rate from $I_3$ to $R_3 = (1 - c_f) \times 1/tI_3$
- $c_f$ - Fatal cases: mortality rate of registered cases
- $d_i$ - Average recovery time in group $R_i$
- $t_i$ - Transition rate from group $R_i$ to $R = d_i^{-1}$
- $tI_1$ - Average time in group $I_1$
- $tI_2$ - Average time in group $I_2$
- $tI_3$ - Average time in group $I_3$

For the simulation process, the rates described above are based on epidemiological studies of COVID-19 that will be described in subsection 3.0.5.

## 3    TUNING WITH GENETIC ALGORITHMS

Several adjustment methods are available to improve the calibration of our model parameters in line with the data collected for each anomalous ICD and ICPC. These methods are frequently used in epidemiological studies and machine learning models. Calibration problems like the ones faced here are commonly addressed through adapted deterministic optimization methods, such as L-BFGS-B Wu et al. (2020b). However, stochastic methods can offer a broader perspective by considering the variety of possible calibration scenarios. Among these stochastic methods, evolutionary genetic algorithms stand out for their effectiveness in solving optimization problems. The next section will explore genetic algorithms' contributions and benefits to our study.

### 3.0.1    DEFINITION

Genetic algorithm (GA) is a bioinspired optimization technique that simulates the process of natural evolution. It operates through mechanisms derived from genetics and natural selection, such as mutation, crossover (recombination), and selection. At its core, GA searches for optimal or suboptimal solutions in complex search spaces using a population of candidate solutions called chromosomes. Each chromosome represents a possible solution and is evaluated based on a fitness function, which determines how well this solution meets the desired objective Goldberg (1989); Mitchell (1998).

### 3.0.2    APPLICATION

The genetic algorithm (GA) is an optimization technique inspired by the natural processes of selection and evolution. In this study, GA was meticulously employed to adjust the parameters of the SEPAI3R3O model, aiming to align the model's predictions with ICD and ICPC data collected from specialized healthcare units.

An initial population of 100 random solutions was generated to start the process. In this context, each solution represents a specific set of parameters $a_i$, $\beta_i$, $P_{sub}$, $I_i$, $E$ of the SEPAI3R3O model,

denoted as $\theta$. The fitness function $f(\theta)$ was defined to evaluate how well the model parameters fit the observed data. It is given by:

$$f(\theta) = \frac{1}{1 + \text{nRMSE}(\theta)} \tag{13}$$

where $\text{nRMSE}(\theta)$ is the normalized mean squared error between model predictions and observed data. This metric quantifies the difference between the values predicted by the model and the actual observed values.

In each of the 100 generations of GA, solutions were evaluated based on their fitness. The ten most suitable solutions were directly selected for the next generation, while the rest were chosen for reproduction using selection mechanisms such as roulette. The probability $p_i$ of a solution $i$ being selected is proportional to its fitness and is given by:

$$p_i = \frac{f(\theta_i)}{\sum_{j=1}^{100} f(\theta_j)} \tag{14}$$

This equation determines the chance of a solution being chosen for reproduction based on its fitness relative to the population's total fitness.

After selection, the solutions underwent crossover (recombination) and mutation operations. The crossover allows the exchange of information between two parental solutions, generating offspring that inherit characteristics from both. During this process, there is a $40\%$ probability of each gene undergoing mutation, introducing random variations in the pre-defined limits for each parameter. This mutation ensures diversity in the population and prevents premature convergence to suboptimal solutions.

The iterative evaluation, selection, crossover, and mutation process was repeated for exactly 100 generations. In each generation, the GA sought to refine the population, getting closer and closer to the optimal solution. The process was completed after the 100th generation. We will now illustrate the proposed model with an example to enhance comprehension.

Initialization: We start with a population of random solutions. Each solution is a set of $\theta$ parameters that will be used in the SEPAI3R3O model. Evaluation: For each solution in the population, we calculate nRMSE by comparing model predictions with observed data. We then use the fitness function $f(\theta)$ to evaluate the quality of each solution. Selection: Based on the calculated aptitudes, we select solutions for reproduction. Solutions with more excellent suitability are more likely to be selected. Crossover and Mutation: The selected solutions are paired and subjected to crossover operations to generate new solutions. Furthermore, there is a $40\%$ chance of mutation in each gene. Iteration: The process is repeated for 100 generations, improving the population at each iteration.

At the end of 100 generations, the solution with the highest fitness is considered the best solution, representing the set of parameters that best fits the observed data. The data used in the adaptation process are the daily accumulated number of patients diagnosed with ICD and ICPC in specialized health units. Therefore, model parameters are expressed in days. The initial conditions for the model are taken from the starting point of the data to be fitted, except for the initial compartments of infected and exposed, which GA determines.

### 3.0.3  COMPUTATION OF THE OPTIMUM GENERATIONS' NUMBER USING CROSS-VALIDATION

Determining the optimal number of generations is crucial to ensure that the Genetic Algorithm (GA) not only converges to a solution but also avoids overfitting. To achieve this, we employ a cross-validation technique Kohavi (1995).

Cross-validation involves splitting the ICD and ICPC data into training and testing sets. The training set is used to tune the parameters of the SEPAI3R3O model, while the test set is used to evaluate the model's performance. By monitoring the error in the test set across GA generations, we can identify the point at which the model starts to overfit the training data. Our approach uses $k-$fold cross-validation Geisser (1975). The data was divided into k subsets of approximately equal size.

In each iteration, $k - 1$ subsets were used for training and the remaining subset for testing. This process was repeated $k$ times, ensuring that each subset was used as a test exactly once. The cross-validation error was calculated as the average of the errors obtained in each $k$ iteration. Observing the evolution of this error over GA generations, we determine the optimal number of generations as the point at which the cross-validation error reaches a minimum. This approach ensures that the GA does not evolve beyond the necessary point, thus avoiding the risk of overfitting the model to the training data and ensuring robust generalization to unseen data James et al. (2013).

### 3.0.4 MODEL VALIDATION

Providing reasonably accurate data, our model successfully reproduces the evolution of SARS in different strata of Recife, for which a sufficient amount of data is available. The strata selected for this analysis were: "0 to 3% ($A$)", "4 to 15% ($B$)" and "16 to 34% ($C$)", which represent the percentage of areas of Communities of Social Interest (CIS) in the neighborhoods Observatório de Saúde Ambiental do Recife (2017).

For each stratum, we focus on the following ICD and ICPCs:

- ICD U07.2: Clinical or epidemiological diagnosis of SARS;
- ICPC R80: Flu;
- ICPC R83: Other respiratory infections.

We chose the training data from when all confirmed cases, recoveries, and death numbers associated with these codes assume non-zero values in the mentioned strata. The active case curve was then projected for the 20 days following that date.

The results obtained by the model when compared to the observed data, showed some discrepancies. These differences can be attributed to the underreporting rate and asymptomatic cases not registered in health systems. As mentioned, the rate of asymptomatic people is $30\%$, leading to a significant increase in cases.

All adjustments to the model presented a coefficient of determination $R^2$ greater than $0.9$, indicating a high correlation between the predicted and observed values. Additionally, because our fitting method is based on a nonlinear regression algorithm, we use a normalized standard error of the estimate to assess the quality of the fits Hyndman & Koehler (2006).

### 3.0.5 EPIDEMIOLOGICAL PARAMETERS

Understanding the dynamics of severe acute respiratory diseases in Recife requires a meticulous analysis of epidemiological parameters due to the pandemic that affected everyone. Therefore, all parameters used in this research are derived from scientific studies of COVID-19, which provided insights into the disease's transmission, progression, and impact on the population.

Like many other regions, the city of Recife faced significant challenges at the start of the pandemic, including underreporting. When the pandemic began manifesting in mid-March, the underreporting rate was $0.50\%$. The underreporting suggests that many cases have yet to be officially recorded or recognized, possibly due to limitations in testing or asymptomatic patients.

Below, we present a table with the main epidemiological parameters related to COVID-19:

These parameters are fundamental for modeling the disease's spread and evaluating interventions' effectiveness. Furthermore, it is crucial to consider local factors, such as population discipline, public health capabilities, and the severity of containment measures, when interpreting and applying these parameters.

## 4 RESULTS AND DISCUSSION

The modeling of the dynamics of COVID-19 in Recife was based on the dynamic model that considers parameters such as transmission rates, disease progression, recovery, and mortality. As described in previous studies, these parameters were backtracked based on measurable quantities, mainly related to COVID-19. The specificity of our analysis lies in the focus on the city's most vulnerable

Table 1: Epidemiological parameters for COVID-19 in Recife

| Parameter | Value | Reference |
|---|---|---|
| Incubation period | 5 days | Linton et al. (2020) |
| Proportion of mild infections | 80% | Yang et al. (2020b) |
| Duration of mild infections | 6 days | Sanche et al. (2020) |
| Proportion of serious infections | 15% | Yang et al. (2020a) |
| Proportion of critical infections | 5% | Zhi (2020)) |
| Case fatality rate | 2% | Russell et al. (2020) |
| $\alpha$ (Recovery rate) | 0.05 | Li et al. (2020) |
| $\beta$ (Transmission rate) | 0.3 | Wu et al. (2020a) |
| $R_0$ (Basic reproduction number) | 2.5-3.0 | Zhou et al. (2020) |

neighborhood strata, considering the heterogeneity of Recife and its division into five strata based on the percentage of areas of Communities of Social Interest (CIS). Rather than relying solely on standard diagnostic tests, our analysis was based on ICD/ICPC data collected at healthcare facilities, providing a more detailed view of the epidemiological situation in the city. This approach, using the SEPAI3R3O model, allowed us to obtain a more accurate representation of the spread of the virus in vulnerable areas of Recife.

The SEPAI3R3O model estimates for the period from April 26, 2020, to March 7, 2021, of COVID-19 in Recife, focusing on the most vulnerable neighborhoods, are in line with the evolution of the pandemic in the city. The estimated primary reproduction number $R_0$ in early March is 3 (95% CI 2.8–3.2). However, by the end of February 2021, this number had reduced considerably to 0.7 (95% CI 0.65–0.75), with an average effective reproduction assessed at 1.6(95% CI 1.5–1.7). This significant decline in $R_0$ reflects efforts to control the pandemic and increased awareness about COVID-19 among the population.

The SEPAI3R3O model estimates for the period from April 26, 2020, to March 7, 2021, of COVID-19 in Recife, focusing on the most vulnerable neighborhoods, are in line with the evolution of the pandemic in the city. The estimated primary reproduction number $R_0$ in early March is 3 (95% CI 2.8–3.2). However, by the end of February 2021, this number had reduced considerably to 0.7 (95% CI 0.65–0.75), with an average effective reproduction assessed at 1.6 (95% CI 1.5–1.7). This significant decline in $R_0$ reflects efforts to control the pandemic and increased awareness about COVID-19 among the population.

Data analysis indicated a notable increase in the protection rate after adopting the first control measures. This rate experienced a jump from 0.0038 at the beginning of the period to 0.0085 between March and April 2020. The average protection rate over the period under analysis was 0.014 (95

The transmission rate increased, driven by the continued spread of the virus and the emergence of outbreaks in densely populated areas of Recife. The average transmission rate was 0.62 (95% CI 0.60–0.64). The mean latent time was 2.8 (95% CI 2.7–2.9) days, while the mean infectious time was 6.1 (95% CI 5.9–6.3) days. On average, the total incubation period was 8.9 (95% CI 8.6–9.2) days.

In contrast to the epidemic parameters of the SEPAI3R3O model, we determined the recovery and mortality rates based on official data provided by the Recife Health Department1 from April 26, 2020, to March 7, 2021−the case data collected from SARS units. The recovery rate ranged between [1.4–3.0%], with an average value of 2.2%. Notably, the mortality rate reduced to around 0.7% at the end of the analyzed period, with an average of 1.2%. The significant decrease in the mortality rate can be attributed to improved available medical care and treatments. The mortality rate showed a stabilization trend in the last few months analyzed. The in-depth analysis of parameter values goes beyond the scope of this work. However, it is essential to highlight that the critical values obtained for Recife align with typical estimates for regions with similar characteristics.

Records of cases related to ICPC R80, R83, and ICD U7.2 were observed during the reference period.

Table 2: Summary of average SEPAI3R3O parameter estimates for the initial phase of COVID-19 in Recife compared to other regions.

| Parameter | Definition | Value for Recife (IC 95%) | Value for other regions (IC 95%) | Reference (other regions) |
|---|---|---|---|---|
| Protection rate (days) | $\alpha$ | 0.017 (0.016–0.019) | 0.086 | Yang et al. (2020b) |
| Transmission rate (days) | $\beta$ | 0.66 (0.64–0.68) | 1.00 | Sanche et al. (2020) |
| Latent time (days) | $\gamma$ | 2.9 (2.8–3.0) | 2.1 | Zhi (2020) |
| Infectious time (days) | $\delta$ | 6.1 (5.9–6.3) | 7.5 | Zhi (2020) |
| Basic reproduction number | $R_0$ | 3.0 (2.8–3.2) | 6.50 (5.73–7.27) | Zhi (2020) |

ICD U7.2 had 16.000 cases, peaked on 2020/11/10, recording 200 cases on that day and a daily average of approximately 44 cases.

ICPC R80 had 4.703 cases, with its highest number of daily cases on 2020/05/22, totaling 108 cases and averaging about 13 cases daily.

Finally, ICPC R83 recorded 2.000 cases, with 2020/05/22 being the day with the highest incidence, 59 cases, and a daily average of 5.5 cases.

The forecast simulations, as shown in Figure 1a, estimated a significant trend in the evolution of COVID-19 in Recife from May 20 to 30, 2020. The simulation, which considers an underreporting rate of 50%, suggests a significant increase in cases. For example, in May 22, when the observed cases totaled 259, the simulation pointed to 388.5 cases, considering underreporting. This simulated value has a 95% confidence interval between 369.075 and 408.925 cases. Additionally, the estimate of asymptomatic cases for the same day was 77.7, with a confidence interval between 73.815 and 81.585. These numbers reinforce the importance of considering underreporting and the presence of asymptomatic cases when assessing the absolute magnitude of the pandemic. Continuous analysis of this data and comparison with simulations are essential to improve the city's disease intervention and control strategies.

Projecting these numbers, we estimate that new infections would decline significantly in mid-July, reaching single-digit numbers in August. However, the total number of active cases would still be above 500 by the end of August. The model predicts that the first wave of the outbreak would extend until October 2020, with an estimated total of 20.000 (95% CI 19.000–21.000) cases, 16.000 (95% CI 15.200–16.800) recovered, and 1.200 (95% CI 15.200–16.800) recovered and 1.200(CI 95% 1.140–1.260) deaths. Understanding that these estimates are derived from current data and may vary as new information becomes available is crucial.

Compartmental models work best with certain population assumptions: a well-mixed group with an equal chance of infection and homogeneity, meaning all react similarly to the disease with consistent transition rules between stages. Any parameters adjusted in the study represent a population average. The model isn't inherently additive, meaning combining results from different regions doesn't equal the outcome for the entire country. Thus, applying the study separately to various cities or regions is advised.

## 5  CONCLUSION

This study presented a novel variation of the SEIR epidemiological model, termed SEPAI3R3O, and employed genetic algorithms to optimize and analyze its associated parameters. The model was meticulously crafted based on socio-demographic data and anomalous ICDs and ICPCs collected from specialized units in Recife, Brazil, from April 26, 2020, to March 7, 2021. Our primary objective was to comprehend the dynamics of disease spread and pinpoint critical factors influencing it. Notably, the underreporting rate, estimated at around 50%, was identified as a significant factor, amplifying the number of cases due to inadequate testing.

Figure 1: Model prediction. (a) Number of cases. The figure shows the peak of the epidemic, corresponding to the maximum number of active cases, occurring between the 20th and 30th of May. (b) Total recoveries and deaths. (c) Time-dependent reproduction number. (d) The number of newly infected individuals per day.

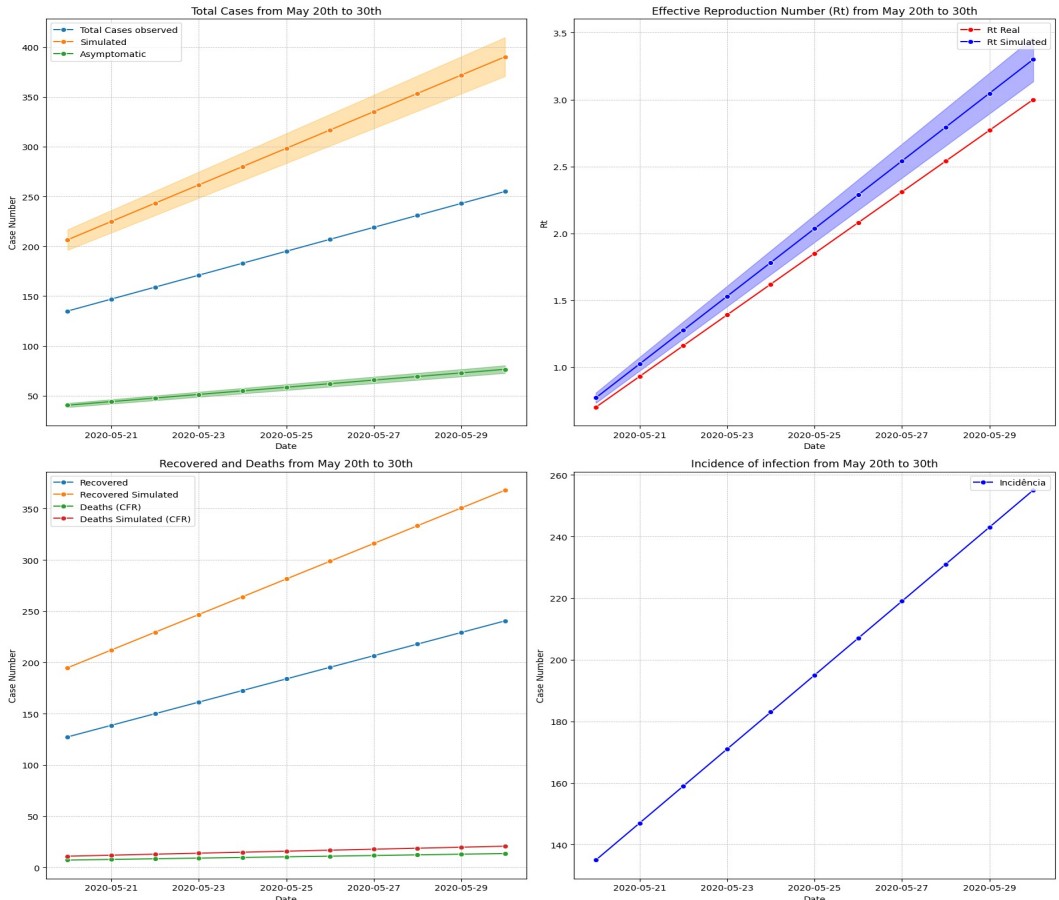

Utilizing the genetic optimization approach, we achieved precise adjustment of the model parameters, leading to enhanced predictions of disease dynamics and a more realistic representation of the number of individuals affected by SARS. The SEPAI3R3O model, when optimized with genetic algorithms, showcased its capability in predicting the disease's spread with an effective reproduction rate $R_0$ of 3 (95% CI 2.8–3.2) and a growth rate of 0.014 (95% CI 0.013–0.015) for the analyzed period. This approach, grounded in realistic data, offers an invaluable tool for researchers and healthcare professionals, aiding in informed decision-making and formulating more effective intervention strategies.

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
