# OpenReview forum: "PARAMETER OPTIMIZATION FOR EPIDEMIOLOGICAL MODEL WITH GENETIC ALGORITHM"
_ICLR.cc/2024/Conference — Submitted to ICLR 2024_

### Official Review · Reviewer_4w3D · 2023-10-31

**Soundness:** 1 poor
**Presentation:** 1 poor
**Contribution:** 1 poor
**Rating:** 1
**Confidence:** 3

**Summary:**

This paper proposes a type of SEIR models that uses genetic algorithms for parameter analysis and optimisation. The dataset studied is based on SARS type diseases (e.g. flu and COVID-19), and consists of sociodemographic and behavioural data from anomalous ICDs and ICPCs, located in Recife, northeast Brazil, from April 26 2020 to March 7 2021. The model is used to study the dynamics of the disease spread and identify the most important factors for the spread of the disease. The use of genetic algorithms is shown to be effective in optimising the parameters of the model, giving better predictive results.

**Strengths:**

- The proposed model crucially considers the problem of underreporting of cases, which is a significant issue in epidemiological modelling.

**Weaknesses:**

- The focus of the paper is not very clear. On the one hand, the paper is about the use of genetic algorithms for parameter optimisation. On the other hand, the experiments section is simply a report of the results of their model. It would be useful to have more discussion on the use of genetic algorithms for parameter optimisation, such as the advantages and disadvantages of this approach, and how it compares to other popular methods.
- The results simply just enumerate what the model outputs, without any intuition or discussion of how good/bad the model is.

**Questions:**

Could this paper be more suitable or impactful in an epidemiology journal? I would like to understand the motivation for submitting to ICLR, such as the benefits that the ML community can bring to this problem, or what the authors hope to gain from bringing this paper to ICLR.

---

### Official Review · Reviewer_yHix · 2023-11-01

**Soundness:** 1 poor
**Presentation:** 2 fair
**Contribution:** 1 poor
**Rating:** 1
**Confidence:** 4

**Summary:**

The authors propose an extended SEIR epidemiological model called SEPAI3R3O for analyzing disease dynamics. Unlike standard SEIR models, they divide each compartment into subpopulations based on detailed characteristics, yielding over 10 parameters. To estimate these parameters, they develop a genetic algorithm optimization approach. The method was applied to model the dynamics of COVID-19 in Recife.

**Strengths:**

1. More detailed modeling of population subgroups.

**Weaknesses:**

1. The proposed SEPAI3R3O model introduces over 10 parameters, which may be difficult to reliably estimate given limited real-world observations. Moreover, modeling these parameters as time-varying further exacerbates the challenge of estimation from scarce data. The authors should discuss the uncertainty in fitted parameter values and provide sensitivity analysis.
2. Limited Novelty. The authors apply a standard genetic algorithm approach for parameter estimation. While computationally fitting an extended SEIR model is non-trivial, the optimization methodology itself does not represent a significant novel contribution. More justification is needed on the novelty of the parameter estimation technique.
3. Unvalidated Estimates. The authors estimate under-reporting rates for infections, but it is unclear how these proposed values were validated. Further analysis should be provided to support the assumed levels of under-reporting across different model compartments.

**Questions:**

What is the observation? How to estimate the Rt and other time-varying parameters?

---

### Official Review · Reviewer_f5YY · 2023-11-10

**Soundness:** 1 poor
**Presentation:** 2 fair
**Contribution:** 1 poor
**Rating:** 3
**Confidence:** 3

**Summary:**

The paper introduces the SEPAI3R3O epidemiological model, an extension of the SEIR model, to analyze and predict the spread of SARS-related diseases, with a focus on COVID-19 in Recife, Brazil. It applies genetic algorithms to optimize the model parameters and make accurate predictions with normalized root mean squared error between model predictions and data as a part of their fitness function. They authors also use k-fold validation during the process of optimizing model parameters. They present results indicating an effective reproduction rate R0 of 3 and a growth rate of 0.014 for the disease using the trained model.

**Strengths:**

- The SEPAI3R3O model incorporates useful extensions like pre-symptomatic transmission and underreporting rate to capture disease nuances.

**Weaknesses:**

- While the application of a genetic algorithm is a legitimate method for parameter optimization, applying it to a new epidemiological model doesn’t itself establish novelty. Genetic algorithms (and other optimization methods) have been widely used in parameter learning/optimization for epidemiological models.
- The paper does not establish the novelty or the necessity of the SEPAI3R3O model modification. There are no experiments comparing the model to existing models. Without a clear distinction of how SEPAI3R3O provides substantial improvements over existing models on well-motivated experiments, the contribution seems incremental. The authors must provide a comparison with other existing models to validate the improvements that SEPAI3R3O claims to offer. This could be done by demonstrating the model's performance against other models on the same dataset or showing how the additional compartments in SEPAI3R3O lead to significantly better understanding or prediction of disease spread.
- The paper is not well-written and missing a lot of details/definitions. At the end of the day, every paper should be self-sufficient to an extent for the reader. The paper doesn’t clearly mention what the data is, how the data is transformed into a format to train the model, what exactly the model is predicting, what the metrics are, and how the metrics are justified, etc. The text should be significantly improved to aid the user in understanding their contribution.
- The paper mentions the adjustment of parameters based on COVID-19 epidemiological studies, it does not specify whether the parameters used were local to the Recife population or if they were generalized from broader studies. Furthermore, the paper estimates an underreporting rate of approximately 50% but does not delve into the methodology for reaching this estimate. Detailed information on the parameter selection process would strengthen the paper's methodological rigor.

**Questions:**

- Can the authors provide comparative analyses with other established models to showcase the advantages of the SEPAI3R3O model in terms of accuracy and predictive capabilities?
- Have you validated the model estimates against other data sources beyond ICD/ICPC codes, like confirmed case counts, hospitalizations, etc.?
- What do the changes in the SEPAI3R3O when compared traditional SEIR models translate into better decision-making for healthcare professionals?
- How was the underreporting rate of 50% estimated, and is this specific to the Recife population or derived from broader data? Could the authors elaborate on the methodology used for this estimation?
- How does the model account for changes in public health policies or behavioral changes in the population over time, and how are these reflected in the model parameters?

---

### Official Review · Reviewer_gtXp · 2023-11-11

**Soundness:** 2 fair
**Presentation:** 1 poor
**Contribution:** 1 poor
**Rating:** 3
**Confidence:** 4

**Summary:**

This study proposes a variation of the SEIR model, termed SEPAI3R3O, optimized with genetic algorithms, focusing on SARS-related data in Recife, Brazil. The proposed approach, combining genetic algorithms with epidemiological modeling, is promising, and using real data is a plus in validating the approach.

**Strengths:**

1) Genetic algorithms to optimize the epidemic model parameters.
2) Using real data to assess the validity of the approach.

**Weaknesses:**

Main Criticisms:
1. Insufficient Literature Review: The paper lacks a broader overview of existing
epidemiological models (see e.g. [2]) and optimization strategies (see e.g. [5-6]). Notably, there is no mention of comparable models like OpenABM [1] or gleam [3]. A section on similar works is essential to evaluate the relevance and novelty of your approach within the field.
2. Lack of Comparative Analysis: Absence of comparison with other optimization algorithms or epidemiological models weakens the paper's impact. Inclusion of comparisons, especially with differential compartmental models and agent-based models, would significantly strengthen the study.
3. Methodology and Reproducibility: The methodology is clear, but including open-source code would greatly enhance reproducibility and scientific verification.
4. Incomplete Results Section: The data presented covers a very narrow time window, which is insufficient to assess the model's predictive accuracy and applicability to broader pandemic dynamics.

Specific Observations:

• Protection Rate Definition: The term "protection rate" is not clearly defined.

• Vague statement: The author writes: "This approach, using the SEPAI3R3O model, allowed
us to obtain a more accurate representation of the spread of the virus in vulnerable areas
of Recife." More accurate with respect to?

• Inconsistencies in Data Presentation: Table 2 shows significantly different values from
other sections without adequate explanation.

• Underreporting Rate. The author writes: "When the pandemic began manifesting in mid-
March, the underreporting rate was 0.50%. " I think it is 50%, isnt'?

• Unended sentence. The sentence: "The average protection rate over the period under
analysis was 0.014 (95 ".

• Simulated Data on Death Rates: The paper should explain why the simulated data shows higher death rates than observed. Usually, the number of deaths is a good indicator of the outgoing epidemic status and usually is not underreported. Do you have an explanation? In this case, the reduction is about 20/30%, from what I can guess from the plot fig.1 plot c.

[1] Hinch, Robert, et al. "OpenABM-Covid19—An agent-based model for non-pharmaceutical interventions against COVID-19 including contact tracing." PLoS computational biology 17.7 (2021): e1009146.

[2] Iranzo, Valeriano, and Saúl Pérez-González. "Epidemiological models and COVID-19: a comparative view." History and Philosophy of the Life Sciences 43.3 (2021): 104.

[3] Gleam Epidemiological Model

[4] Lau, Hien, et al. "Evaluating the massive underreporting and undertesting of COVID-19 cases in multiple global epicenters." Pulmonology 27.2 (2021): 110-115.

[5] Gopal, K., Lee, L. S., & Seow, H. (2020). Parameter Estimation of Compartmental Epidemiological Model Using Harmony Search Algorithm and Its Variants. Applied Sciences, 11(3), 1138.

[6] Bent, O. E., Wachira, C., Remy, S. L., Ogallo, W., & Walcott-Bryant, A. (2021). A Framework for Inferring Epidemiological Model Parameters using Bayesian Nonparametrics. AMIA Annual Symposium Proceedings, 2021, 217-226.

**Questions:**

Improvement Suggestions:
1. Enhance Literature Review: Expand the literature review to include a comprehensive
comparison with existing epidemiological models and optimization strategies, highlighting
the uniqueness and advantages of your approach.
2. Broaden Results Section: Extend the time window in the results section to provide a more
thorough evaluation of the model's predictive capabilities.
3. Proofreading and Editing: Ensure the paper is carefully proofread to avoid repetition and
incomplete sentences.
4. Comparative Analysis with Established Techniques: Discuss how your approach to inferring
the underreporting rate compares with simpler, established techniques, providing a clear justification for the chosen methodology.

---

### Meta-Review · Area_Chair_RN85 · 2023-12-05

**Metareview:**

The reviewers did not find this paper strong enough and suitable for the conference.

**Justification For Why Not Higher Score:**

All the reviewers were confidently negative about the suitability of this submission to the conference.

**Justification For Why Not Lower Score:**

N/A

---

### Decision · Program_Chairs · 2024-01-16

Reject